# A comprehensive survey of cancer medicines prices, availability and affordability in Ghana

**Phyllis Ocran Mattila**[1]*, **Richard Berko Biritwum**[2], **Zaheer Ud-Din Babar**[1]

**1** Department of Pharmacy, University of Huddersfield, Huddersfield, United Kingdom, **2** Department of Community Health, University of Ghana Medical School, Accra, Ghana

* phyllis.ocran@hud.ac.uk

## Abstract

### Introduction

In Ghana, prices for cancer medicines are characterized by high retail markups, forex fluctuations and high variation in prices of medicines. Most patients cannot afford the cancer medicines. There is a problem of unaffordability and limited availability of essential cancer medicines which suggests potential inequity in patient access to cancer medicines. The study objective was to assess the prices, availability, and affordability of cancer medicines in Ghana. Prices of cancer medicines are a major contributor to the cost of treatment for cancer patients and the comparison of these cost was assessed to determine the affordability.

### Method

The methods developed and standardized by the World Health Organization (WHO) in collaboration with the Health Action International (HAI), was adapted and used to measure prices, availability, and affordability of cancer medicines in Ghana. The availability of cancer medicines was assessed as percentage of health facilities stocked with listed medicines. The price of cancer medicines (of different brands as well as the same medicine manufactured by different pharmaceutical industries) available in the public hospitals, private hospitals, and private pharmacies was assessed, and the percentage variation in prices was calculated. Medicine prices were compared with the Management Sciences Health's International Reference Prices to obtain a Median Price Ratio (MPR). The affordability of cancer medicines was determined using the treatment cost of a course of therapy for cancer conditions in comparison with the daily wage of the unskilled Lowest-Paid Government Worker.

### Results

Overall availability of cancer medicines was very low. The availability of Lowest Priced Generic (LPG) in public hospitals, private hospitals, and private pharmacies was 46%, 22%, and 74% respectively. The availability of Originator Brand (OB) in public hospitals, private hospitals, and private pharmacies was 14%, 11%, and 23% respectively. The lowest median price [United States Dollars (USD)] for the LPG was 0.25, and the highest median price was 227.98. For the OB, the lowest median price was 0.41 and the highest median price was

**Data Availability Statement:** All relevant data are within the paper and its Supporting Information files.

**Funding:** The author(s) received no specific funding for this work.

**Competing interests:** The authors have declared that no competing interests exist.

**Abbreviations:** WHO, World Health Organization; HAI, Health Action International; MPR, Median Price Ratio; LPG, Lowest Priced Generic; OB, Originator Brand; USD, United States Dollars; LMICs, Low- and Middle-Income Countries; EML, Essential Medicines List; NEML, National Essential Medicines List; MOH, Ministry of Health; GHS, Ghana Health Services; NHIS, National Health Insurance Scheme; NMP, National Medicines Policy; STG, Standard Treatment Guidelines; EMLc, Essential Medicines list for Children; INN, International Nonproprietary Name; Gh, Ghana; MUP, Median Unit Price; MSH, Management Sciences for Health; IRP, International Reference Price; CPI, Consumer Price Indices; HPM, Highest-Priced Medicine; LPM, Lowest-Priced Medicine; PR, Price Ratio; LPGW, Lowest-Paid Government Worker; emc, Electronic Medicines Compendium; UK, United Kingdom; NCCN, National Comprehensive Cancer Network; ERC, Ethics Review Committee; GHS, Ghana Health Service; KBTH, Korle Bu Teaching Hospital; KATH, Komfo Anokye Teaching Hospital; GoG, Government of Ghana; SEP, Single Exit Price; ERP, External Reference Pricing.

1321.60. The lowest and highest adjusted MPRs of OBs and LPGs was 0.01 and 10.15 respectively. Some prices were 20.60 times more expensive. Affordability calculations showed that patients with colorectal and multiple myeloma cancer would need 2554 days wages (5286.40 USD) and 1642 days wages (3399.82 USD) respectively to afford treatment.

## Conclusion

The availability of cancer medicines was very low, and less than the WHO target of 80%. There were considerable variations in the prices of different brands of cancer medicines, and affordability remains suboptimal, as most patients cannot afford the cancer medicines. Comprehensive policies, regulations and multifaceted interventions that provides tax incentives, health insurance, and use of generics to improve cancer medicines availability, prices, and affordability, for the masses should be developed and implemented in Ghana.

## Introduction

Cancer is a great public health issue. The estimated global cancer burden was 19.3 million new cases and 9.9 million deaths in 2020. This may increase to 30.2 million new cases by 2040 [1]. In Africa, new cancer cases were 1.1 million in 2020 and is projected to reach 2.1 in 2040 [1].

In 2020, Ghana had 24,009 cancer cases (14,078 in females, 9,931 in males) and 15,089 deaths, with new cases projected to be 44,475 (25,518 in females, 18,957 in males) in 2040 [1]. The most common cancers in adults are breast (18.7%), liver (14.4%), cervix uterus (11.6%), prostate (8.9%), non-Hodgkin's lymphoma (5%), ovary (4.2%), Colo rectum (3.3%), and other cancers (33.9%). And for children; non-Hodgkin's lymphoma (20%), leukemia (16.1%), kidney (11%), BNS (7.2%), liver (2.2%), Hodgkin's lymphoma (1.9%), naso pharynx (1.7%) and other cancers (40%) [1].

The cancer burden and mortality can be reduced with early diagnosis and treatment [2]. This depends on the equitable access to available and affordable cancer medicines. In Low- and Middle-Income Countries (LMICs), health system strengthening can improve various facets of the medicine supply chain including access, affordability, and quality of medicines [3].

Over 20% of African countries have no access to cancer medicines at all, while access is limited and sporadic in other countries [4]. There are concerns about the lack of adequate access to both new and off-patent cancer medicines, with soaring prices cited as a main contributory factor [5].

In 2018, a WHO study found that, pricing of cancer medicines was disproportionately higher than other types of pharmaceuticals and therapies. Prices increased in the absence of regulations and non-uniform pricing policies, resulting in inefficient cost-shifting activities and potential inequity. Thus, a greater level of price control can lower prices and improve access [5].

Many cancer patients cannot afford the cost of treatment, especially if they lack an adequate insurance coverage, government reimbursement or funding [5]. Patients with inadequate insurance coverage often suffered financial hardships because of the prohibitive cost of cancer medicines, to the extent that they may lower the treatment dose, partially fill prescriptions, or even forego treatment altogether [5].

Most cancer patients do not have affordable access to treatment because of low availability of cancer medicines. Some reasons include the absence of the cancer medicines on the Essential Medicines List (EML), inefficient supply chains, unreliable or lack of suppliers, and budgetary constraints [5–7]. Studies showed that in LMICs and in the 37 African countries subscribing to the WHO EML, the number of cancer medicines listed on the National Essential Medicines List (NEML) was eighteen and thirteen respectively, with many not listed [5, 8]. A WHO report showed that countries especially Africa with lower national income had lower availability of cancer medicines, or availability only with higher out-of-pocket patient payments, especially for high-cost medicines, including targeted therapies [5].

In Ghana, health care is provided by the government and largely administered by the Ministry of Health (MOH) and Ghana Health Services (GHS). There are about 29 facilities including government specialist/referral hospitals; Korle Teaching Hospital, Komfo Anokye Teaching Hospital, Cape Coast Teaching Hospital, Tamale Teaching Hospital, the 37 Military Specialist Hospital, private hospitals, and pharmacies, that provides cancer medicines to patients [9].

All cancer medicines are imported from abroad by private licensed medicine suppliers/wholesalers, so the prices are largely affected by forex fluctuations, as well as import tariffs and duties, which together with the profit markup cost, renders it very expensive and unaffordable to most patients [10]. Many cancer medicine purchases are small in volume and, therefore, high in price. To promote universal coverage and equity in healthcare delivery services, a National Health Insurance Scheme (NHIS) was set up in 2003 to enable Ghanaians make yearly contributions into a fund so that in the event of illness, they can be supported by the fund to receive affordable health care. Ghanaian contributors are grouped according to their levels of income and premiums are based on the capacity to pay [11]. Only cervical and breast cancer treatments are funded by the NHIS, thus, majority of cancer medicine's costs are made through out- of- pocket payments popularly known as 'cash and carry' [12]. With the minimum daily wage being approximately $2, the out- of- pocket cost of cancer care is very expensive and borne with much difficulty, pushing many into poverty [7]. This compels most patients/guardians to seek healthcare from herbalists or spiritual healers leading to a delay in diagnosis and treatment or default in treatment [12].

Price, availability, and affordability are the main criteria to measure whether patients can purchase medicines at an affordable price. Price transparency when enshrined in the National Medicines Policy (NMP), will help strengthen government's negotiating position and enhance their ability to obtain more affordable medicines [13]. The NMP, NEML and Ghana Standard Treatment Guidelines (STG), includes a pricing policy to improve pricing mechanisms and promote affordability of medicines [14–16]. The impact on affordability though is unclear.

Several studies have examined the availability, price, and affordability of medicines [7, 17–30], however none has been done to show the landscape of adults' cancer medicines pricing, availability, and affordability in Ghana. This study, which to the best of our knowledge, is the first national comprehensive survey of adults' cancer medicines pricing, availability, and affordability in Ghana. A major barrier to creating and implementing national childhood cancer strategies is the limited data on the cost of delivering childhood cancer medicines [31]. Thus, this survey also assessed the pediatric cancer medicines pricing, availability, and affordability.

This study will add to scientific literature and improve the knowledge and equity considerations regarding pricing, affordability, availability, and access to cancer medicines. And thus, contribute to the development of pricing models, policies, and strategies to improve the survival and quality of life of the cancer patients. This study aims to assess the variation in prices

of different brands of cancer medicines, its availability and affordability in the Private Hospitals, Public Hospitals, and Private Pharmacies of Ghana.

## Methods

### Sampling strategy

The study was conducted to examine the pricing, availability, affordability, and access to cancer medicines in Ghana, using an abridged WHO/HAI methodology [32]. A total of 29 facilities [7 public (teaching/tertiary/referral) hospitals, 20 private pharmacies and 2 private hospitals] in 4 survey areas (Greater Accra, Ashanti, Central and Northern Regions respectively), were purposively surveyed [9]. The facilities were selected because they were the only facilities in Ghana providing cancer medicines to patients during the survey period. This constitutes a national survey, as they are representative of the whole country. Each survey area covered a population of about 1.9 million to 5.9 million and reachable within one day's travel from Ghana 's main capital city, Accra [33] (See Table 1).

### Survey medicines

According to the requirements of the WHO/HAI methodology, surveyed medicines were selected based on the local patients' disease burden and needs. Data was collected on Sixty-five (65) cancer medicines (with different strengths and dosage form). The selection was based on cancer medicines listed in the 2017 Ghana NEML, 2021 WHO EML, 2021 WHO Essential Medicines list for Children (EMLc), and medicines used for cancer treatment in Ghana, due to its availability, utilization, and the burden of disease in Ghana [14, 34, 35] (See S1 Table).

In this study, the term cancer medicines refer to cytotoxic and adjuvant medicines. The cytotoxic medicines include alkylating agents, antimetabolite analogs of folic acid, pyrimidine, and purine, natural products, hormones and hormone antagonists, and a variety of agents directed at specific molecular targets were assessed [36].

### Data collection

A cross sectional study was conducted from August 2020 to October 2020. Trained research assistants visited the selected retail and hospital pharmacies and filled a data collection sheet/questionnaire designed by the authors with the support of the pharmacist on duty. The questionnaire was adapted from the WHO/HAI methodology and had 3 sub sections on demographics (the medicine outlet, etc.), medicine's availability, and unit prices [32]. It was pilot tested and used for the data collection. The patient prices were collected for all cancer medicines found in the facility on the day of the survey. For quality control, the research assistants checked the completeness and consistency of the data collected at the end of each day. The data collection form is available from the authors upon request.

**Table 1. Number of facilities (outlets sampled) and geographical location of participating cities [33].**

| Region | Regional Population (2020) [33] | City Surveyed | Number of Facilities in each City | | |
|---|---|---|---|---|---|
| | | | Public Hospitals | Private Pharmacies | Private Hospitals |
| Ashanti | 5,924,498 | Kumasi | 1 | 7 | 0 |
| Greater Accra | 5,055,883 | Accra | 4 | 6 | 1 |
| Central | 2,605,492 | Cape Coast | 1 | 2 | 1 |
| Northern | 1,948,413 | Tamale | 1 | 5 | 0 |
| Total Facilities | | | 7 | 20 | 2 |

The Data was double checked for completeness and accuracy with blanks and errors corrected and entered on a standardized computerized workbook and Microsoft excel spread sheet for analysis. In every facility, data was collected on availability and the unit price (price of pack divided by pack size i.e., the price per cap, tab, ml, dose, gram), of the LPG and the OB. The OB is the original pharmaceutical product that was first authorized for marketing. The International Nonproprietary Name (INN)/ (generic name) refers to products other than the OB that contains the same active ingredient and marketed under a generic name. The LPG is the lowest price of generic products found at each medicine outlet (place where medicines are dispensed to patients) [32].

Data was collected in local currency, Ghana (Gh) Cedis and converted to USD using foreign exchange rate of (1 Gh Cedis = 0.1652 USD =), as of 06 October 2021 [37].

## Analysis

For statistical analysis, the cancer medicines found in all 3 sectors were included. Each medicine's Median Unit Price (MUP) was calculated instead of the mean values. The MUP is the median procurement price per unit dose [32, 38]. The 'mean-median' among the 3 sectors for matched sets of medicines was compared and the 'mean-median' and standard deviations in prices, availability, and affordability were presented. A Kruskal-Wallis test was applied and $p < 0.05$ was used to indicate a significant difference.

**Availability.** The availability of cancer medicines was determined by a physical inspection of the medicine outlets having cancer medicines (OB and LPG) on the day of data collection.

The percentage (%) availability of each medicine at each survey site was calculated as:

[(Physical availability of medicine at hospital/private pharmacies) divided by (Number of hospital/private pharmacies)] X 100.

Each medicine was also reported as being listed or not listed on the NEML, WHO EML and EMLc.

**Price.** The 2015 Management Sciences Health (MSH), Supplier International Reference Price (IRP) was used to compare local cancer medicine prices to an international standard [39]. The IRPs are the medians of recent procurement or tender prices offered by predominantly not-for-profit suppliers to developing countries for multi-source products [39]. The MPR is the ratio of median local unit price of a surveyed medicine to its IRP, and was calculated as follows:

MPR = Median local unit price (USD)/Median IRP (USD).

For this study, the MPR was calculated using prices having a deflation factor of 84.73% according to the Ghana Consumer Price Indices (CPI) for July 2014 to July 2019 [40], as the latest available IRPs were published in 2015 [39]. Like other studies, this price adjustment method was adopted to make the comparison between local and international prices more reliable [38, 41, 42]. Normally, an MPR of one or less is taken as efficient procurement in the public sector, while below three, it is considered efficient for the private sector [32, 43].

The retail price of each unit of the individual medicine with the same strength and dosage form from all manufacturers was determined, and in each facility if more than one price was found, the lowest price was taken. All facilities patient retail data was combined to determine for each medicine (OB and LPG), its lowest price, highest price, and MUP, including if only one price was found.

The difference between the Highest-Priced Medicine (HPM) and Lowest-Priced Medicine (LPM), and Brand premiums between the OB and their LPG equivalents was determined [32].

The difference in the maximum and minimum price of the individual medicine being manufactured by several pharmaceutical industries across different brands was calculated using the

following formula: *Price Variation/Cost Differential (%) = [(Price of the Originator/Brand with highest price–Price of the Generic/Brand with lowest price) divided by (Price of Originator/Brand with highest price)] x 100*.

The Price Ratio (PR) between OB and LPG or HPM and LPM was calculated as:

*Price Ratio = Price of the OB divided by Price of the LPG or Price of the HPM divided by Price of the LPM*. If the calculated ratio is ≤1 the reported price is reasonable in the public sector. A value ≥3 in the private sector means Ghanaians pay more for the medicine than recommended by WHO [32].

**Affordability.** The affordability of treating key cancer health problems using standardized treatment regimens was calculated using the median prices collected during the survey. Based on the WHO/HAI method, affordability was measured by calculating the cost of a month supply of medicines for the duration of cancer therapy, using daily dose and course of treatment in comparison with the daily wage of the Lowest Paid Government Worker (LPGW) [32]. In this study, treatment regimens from the Electronic Medicines Compendium (emc) [44], United Kingdom (UK) and the National Comprehensive Cancer Network (NCCN) treatment guidelines [45] were used. A month of cancer medicines was used to demonstrate the economic implication on a patient if they would have to pay for it out of pocket, even though a cancer patient is expected to have more than one cycle of treatment with multiple medicine regimens. Calculations were based on an 80 kg adult where needed. The 2021 salary of the LPGW in Ghana is 12.53 Gh Cedis (2.07 USD) per day [10].

Number of days wage to afford treatment = *cost of tablet(s)/vial(s) of cancer medicine needed per month /daily wage of lowest paid government worker*. The WHO recommends that for a treatment to be affordable it should not exceed 1 day's wages using the wage of the unskilled LPGW [32].

It is important to bear in mind that these costs refer only to the medicine component of the total treatment costs. Consultation fees and diagnostic tests were not considered.

## Ethical considerations

No cancer patient/human subject (adult and minor) was selected nor interviewed in this study, as the study involved a collection of data on the pricing and availability of cancer medicines. Ethical approval for the study was granted by the University of Huddersfield, UK (Ref: SAS-SREIC 19.11.19–2); Korle Bu Hospital technical committee (Ref: KBTH-STC 00003/2020), Ghana Health Service technical committee (Ref: GHS-ERC 007/01/20), and the Komfo Anokye Teaching Hospital Institutional Review Board Research & Development Unit (Ref: KATH IRB/CA/102/20), respectively. All ethical principles and guidelines were adhered to throughout the study and all the ethical approval letters were shared with the managers of the medicine's outlets. Administrative permission and informed consent were obtained and granted by phone (verbally) by the managers of the medicine's outlets prior to data collection.

Due to the COVID 19 pandemic, safety aspects that adheres to national directives on social distancing, washing of hands with soap and water, use of hand sanitizers and use of face masks were followed.

## Results

Three broad sectors; public (public hospitals), private (private pharmacies) and other (private hospitals) were analyzed for the survey (See Table 1).

Sixty-five (65) individual cancer medicines (LPGs and OBs of different strengths and dosage form) were surveyed in 29 medicine outlets (See S1 Table).

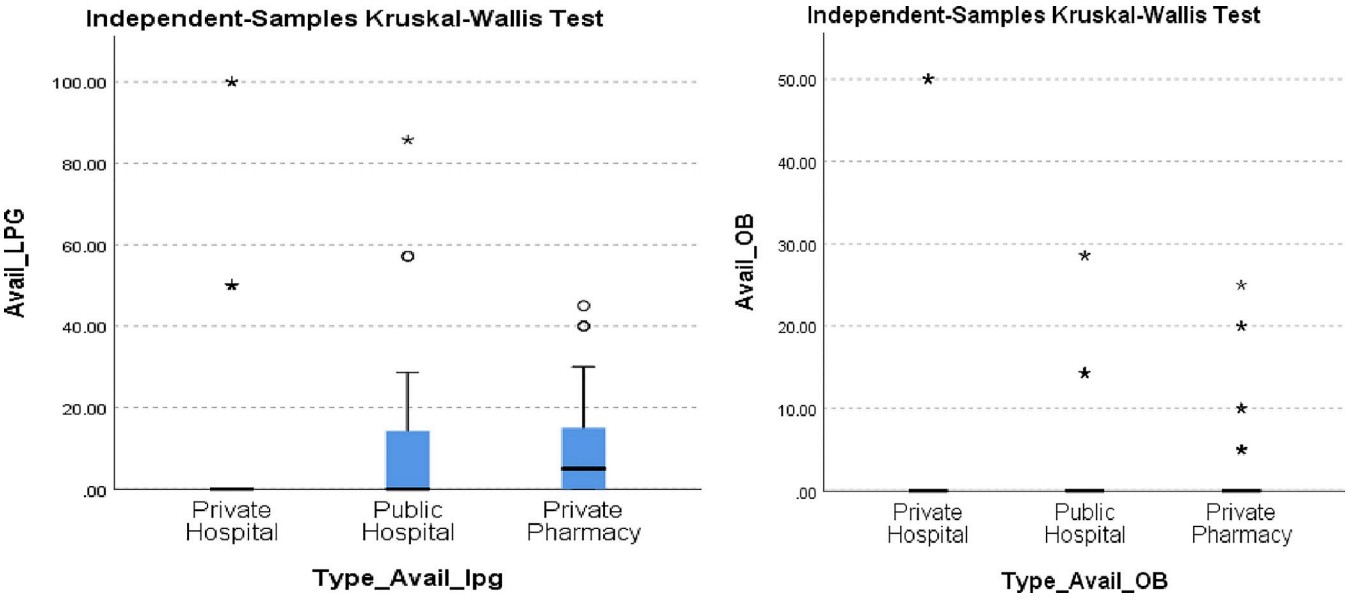

**Fig 1.** a and b. Percentage availability of LPG and OB using Kruskal Wallis Test.

## Availability

Overall availability of the cancer medicines (OB and LPG) in all 29 survey facilities was very low. The comparisons between medicine groups and facilities were not statistically significant as p>0.05. The LPGs were more available than the OBs in all sectors. The percentage availability of LPG And OB medicines using Kruskal Wallis test is shown in Fig 1A and 1B respectively.

The 'mean of medians' availability of the LPGs was highest in the private hospitals (13.08%), then public hospitals (10.55%), and the private pharmacies (9.69%) had the lowest availability (2.42%). (See Table 2, Fig 1A). The 'mean of medians 'availability of the OBs was more in the private hospitals (5.38%) than in the private pharmacies (2.46%), with the lowest being in the public hospitals (2.42%) (See Table 2, Fig 1B).

The availability of LPGs and OBs in public hospitals was 46% and 14% respectively (See Fig 2A), and the availability of LPGs and OBs in private hospitals was 22% and 11% respectively (see Fig 2B). In the private pharmacies, the LPGs and OBs availability were 74% and 23% respectively (See Fig 2C).

16.92 percent of cancer medicines found in the medicine outlets were listed on the NEML. There were other strengths listed on the NEML, but not available to be surveyed. Absence of a medicine from the NEML did not necessarily mean it was unavailable, as 83.08% of cancer medicines found in the medicine outlets were not listed on the NEML. The WHO EML and WHO EMLc listed 66.15% of the cancer medicines.

**Table 2. Percentage availability data using 'mean of median'.**

| Facility | OB Mean | LPG Mean |
|---|---|---|
| Private Hospital | 5.38 | 13.08 |
| Public Hospital | 2.42 | 10.55 |
| Private Pharmacy | 2.46 | 9.69 |
| All Facilities | 3.42 | 11.11 |

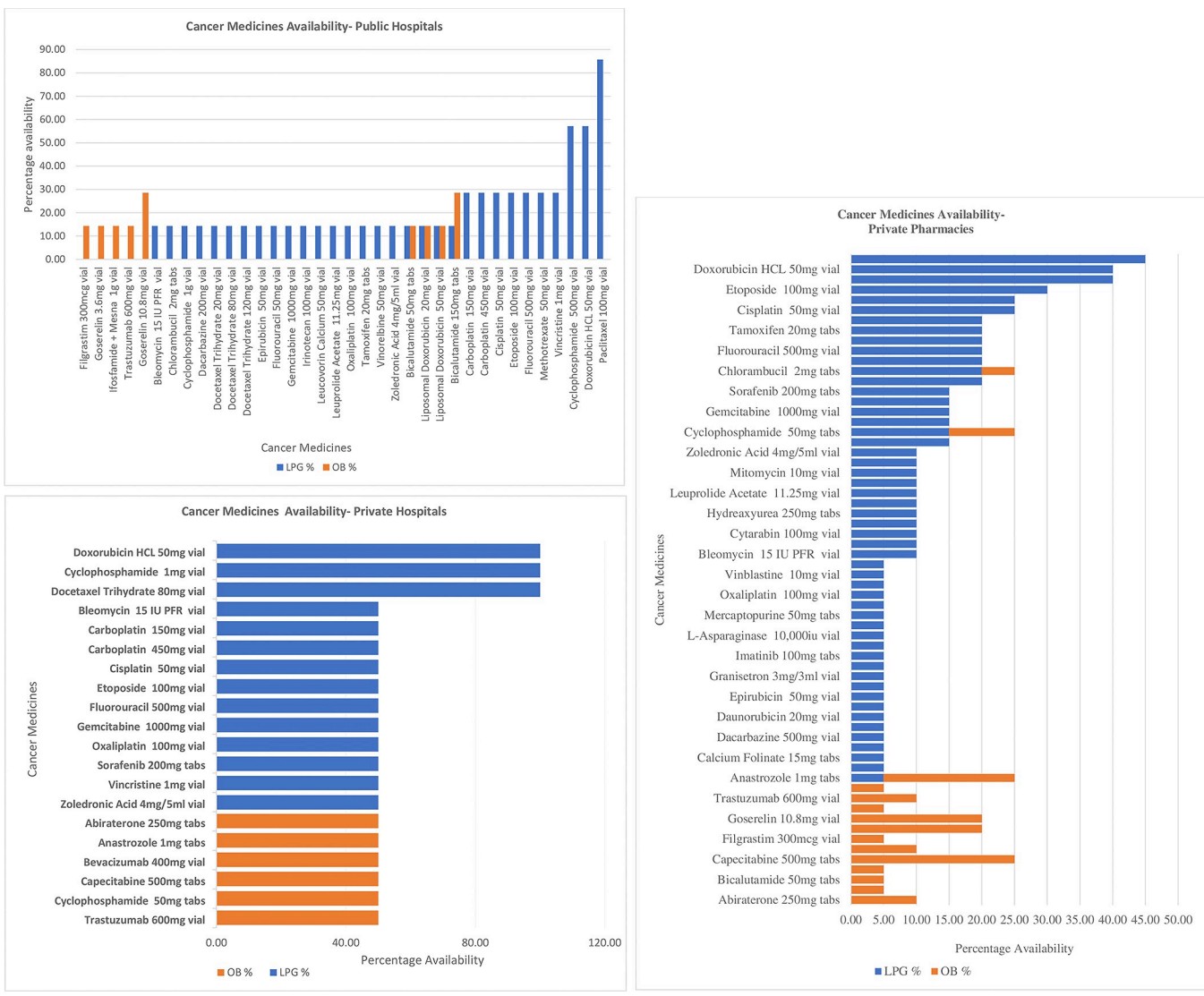

**Fig 2.** a. Availability of Cancer Medicines (OB and LPG) in Public Hospitals. b. Availability of Cancer Medicines (OB and LPG) in Private Hospitals. c. Availability of Cancer Medicines (OB and LPG) in Private Pharmacies.

The availability of Cancer Medicines (OB and LPG) in Public Hospitals, Private Hospitals, and Private Pharmacies is shown in Fig 2A–2C respectively.

## Price ratio comparisons

The 65 sampled cancer medicines (OB and LPG) showed significant price differences between their HPM and LPM (See Table 3). The price comparisons were not statistically significant as $p > 0.05$. The median price variations between OB and LPG using Kruskal Wallis Test are shown in Fig 3A and 3B below.

The mean of median prices (USD) of the LPGs was highest in the private hospitals (55.42), followed by the public hospitals (32.10) with the lowest being the private pharmacies (31.99). The prices ranged from a minimum of 0.25 to a maximum of 227.98. The highest median price was found in private pharmacies (227.98), private hospitals (165.20) and the lowest in public hospitals (132.16) (See Table 3, Fig 3A and 3B).

**Table 3. Median price (USD) variations between HPM, LPM, OB, LPG.**

| Facility Price | Mean of Median LPG | Minimum of Median LPG | Maximum of Median LPG | Mean of Median OB | Minimum of Median OB | Maximum of Median OB |
|---|---|---|---|---|---|---|
| Private Hospital | 55.42 | 2.97 | 165.20 | 391.39 | 0.41 | 1321.60 |
| Public Hospital | 32.10 | 0.48 | 132.16 | 120.19 | 1.43 | 646.59 |
| Private Pharmacy | 31.99 | 0.25 | 227.98 | 104.67 | 0.50 | 581.21 |
| All facilities | 35.59 | 0.25 | 227.98 | 169.72 | 0.41 | 1321.60 |

The mean of median prices (USD) of the OBs was highest in the private hospitals (391.39), then public hospitals (120.19) with the lowest being the private pharmacies (104.67). The median prices ranged from a minimum of 0.41 to a maximum of 1321.60. The highest median price was found in private hospitals (1321.60), public hospitals (646.59) and the lowest in private pharmacies (581.21) (See Table 3, Fig 3A and 3B).

In the public hospitals, 8.33% of the cancer medicine's price differential was above 50%, with a cost differential ratio higher than 2. Bicalutamide 150 mg tablets (95.15%) and Vincristine 1mg vial (5.56%) had the highest and lowest price differentials respectively (See S2 Table). The private hospitals showed that 66.67% of their medicines had cost differentials above 50%, and ratios higher than 2. Doxorubicin HCL 50mg vial (88.75%) and Docetaxel Trihydrate 80mg vial (21.67%) had the highest and lowest cost differential respectively (See S3 Table). In the private pharmacies, 77.78% had a price ratio <3 and 33.33% of the cancer medicines had a price variation ≥50%. Thalidomide 50 mg capsules (97.92%) had the highest price differential and Leuprolide acetate 11.25mg vial had the lowest price differential (4.27%) (See S4 Table).

Fig 4A and 4B shows the median price variability/cost differences between 6 OBs and 6 LPG from public hospitals, private hospitals, and private pharmacies. Only those medicines for which both the originator brand and a generically equivalent product were found, were included in the analysis to allow for the comparison of prices between the two product types.

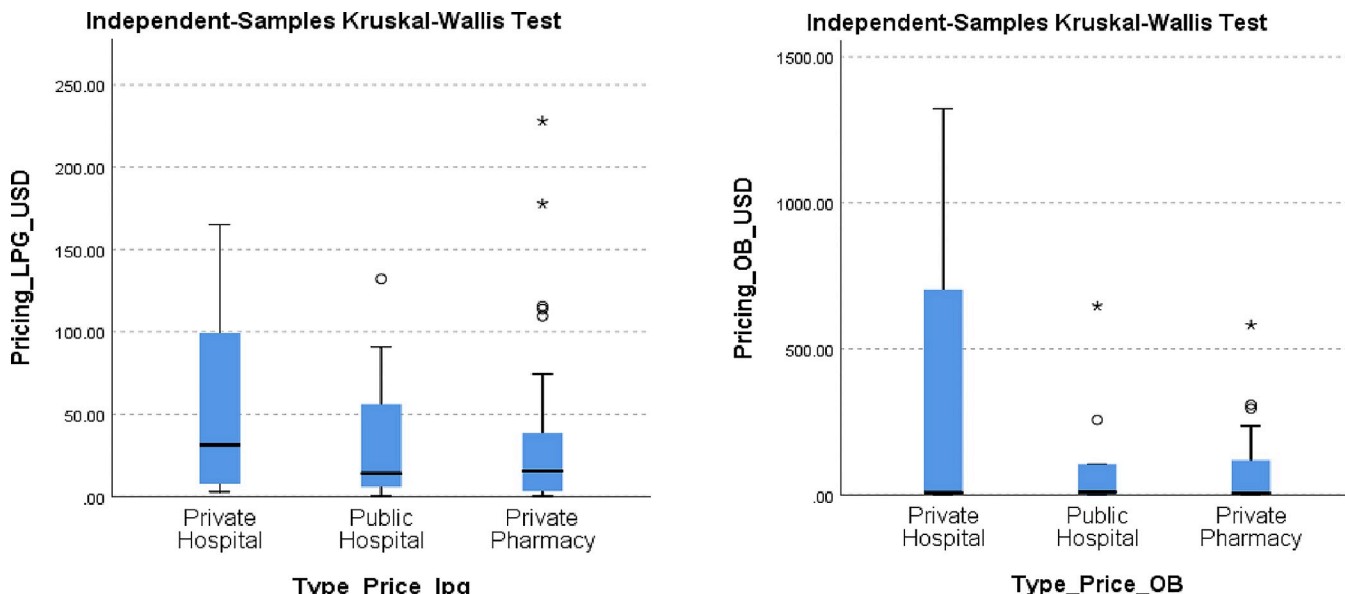

**Fig 3.** a and b. Median Price Variations between OB and LPG using Kruskal Wallis Test.

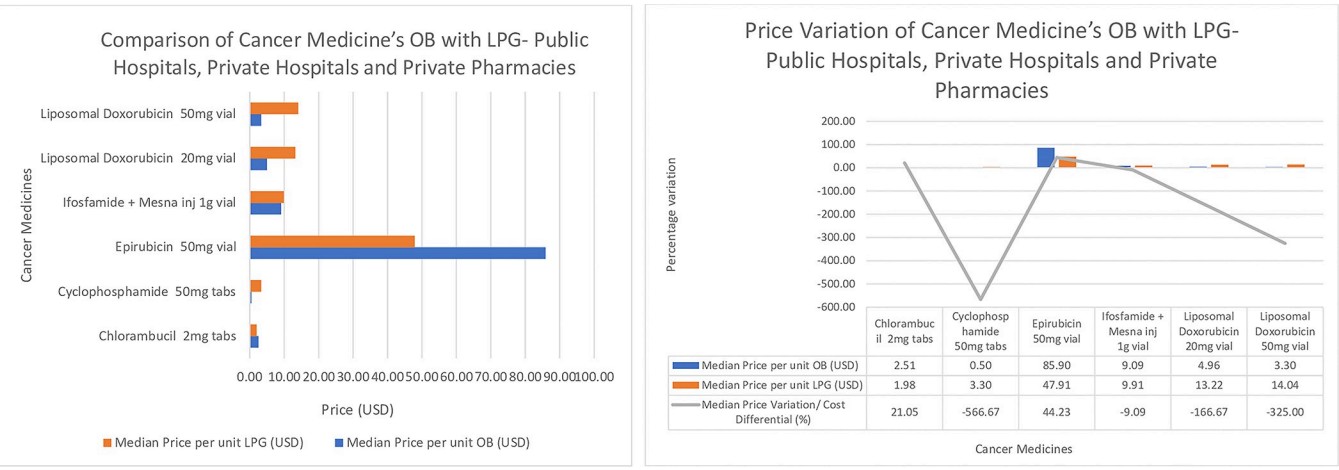

**Fig 4.** a. Comparison of Cancer Medicine's OB with LPG- Public Hospitals, Private Hospitals and Private Pharmacies. b. Price Variation of Cancer Medicine's OB with LPG- Public Hospitals, Private Hospitals and Private Pharmacies.

The lowest OB and LPG price variation was chlorambucil 2mg tablet (21.05%) and the highest was Epirubicin 50mg vial (44.23%) respectively, depicting that some OBs were more expensive than the LPGs. Cyclophosphamide 50mg tablets showed the highest negative price variation of -566.67%. Liposomal Doxorubicin 50mg vial, Liposomal Doxorubicin 20mg vial, Epirubicin 50mg vial showed negative price variations of -325.00%, -166.67% and -9.09% respectively, as these LPGs, were more expensive when compared to the OBs.

The MPR analysis included all medicines (OBs and LPGs) with IRPs. The inflation adjusted MPRs across Public Hospitals for OBs had a low value of 0.01 and a high value of 7.71, whilst the LPGs had a low value of 0.08 and a high value of 10.15. The inflation adjusted MPRs across Private hospitals for OBs had a low value of 0.21 and the highest value was 0.61, whilst the LPGs had a low value of 0.14 and a high value of 2.41. The inflation adjusted MPRs across Private Pharmacies for OBs had a low value of 0.12 and the highest value was 1.03, whilst the LPGs had a low value of 0.03 and a high value of 1.66 (See S5–S7 Tables). Only 10.34%, 11.76% and 8.70% of medicines in Public Hospitals, Private Hospitals and Private Pharmacies respectively had an MPR more than 1.

## Affordability

Affordability was assessed for all the cancer medicines (OB and LPG) in comparison to the daily wage of the unskilled LPGW and found to cost more than 1 day's wage. There were differences in affordability in all the sectors, but they were not statistically significant as p>0.05.

The affordability of OB and LPG using Kruskal Wallis Test is shown in Fig 5A and 5B. When comparing sectors, cancer medicines are more affordable in public hospitals (OB-188.00 and LPG-54.81). The private hospitals (OB-222.06 and LPG-105.64) are more affordable than the private pharmacies, which are the most expensive (OB-653.57 and LPG-134.83). (See Table 4 and Fig 5A and 5B).

In all sectors, one-month treatment with the most expensive medicines requiring several days wages were included. Bevacizumab 400mg vial (OB) required 2554 days' wages and Thalidomide 50mg cap (LPG) required 1642 days wages and were the most expensive and unaffordable cancer medicines.

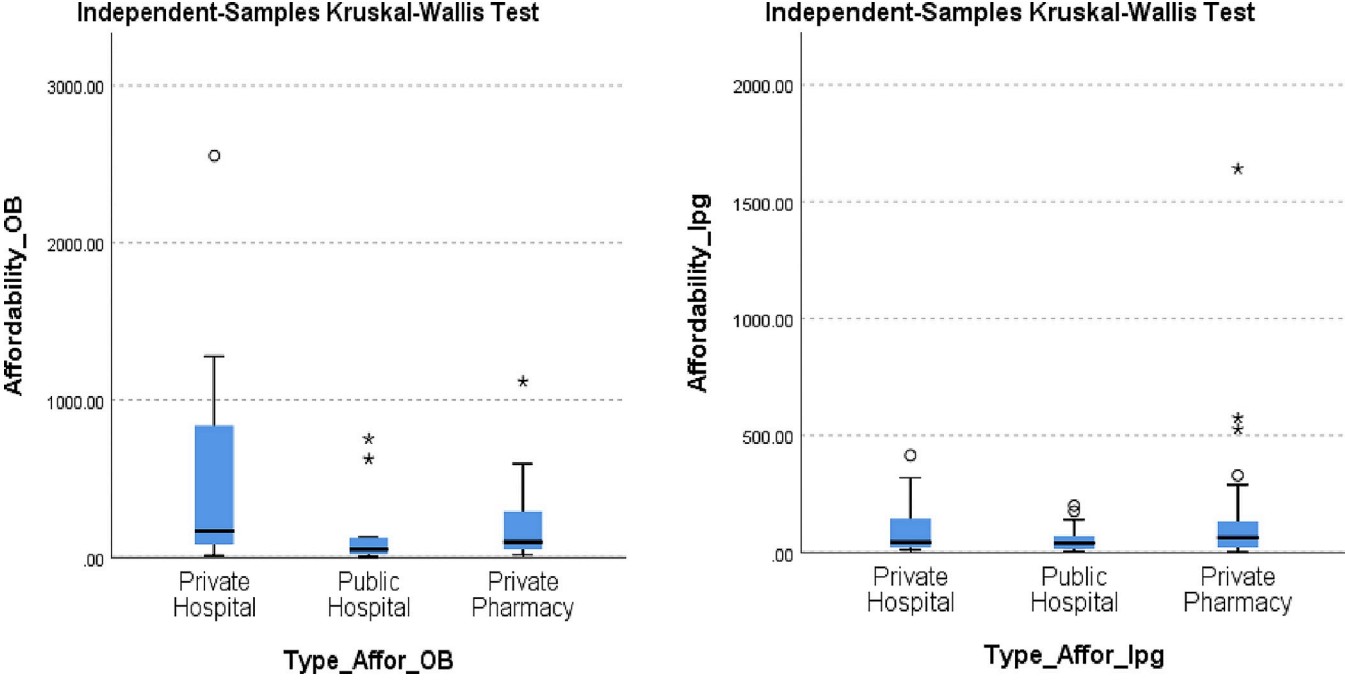

**Fig 5.** a and b: Affordability of OB and LPG using Kruskal Wallis Test.

In all sectors, all treatments with OBs, costs more than 1 day's wage. In the public hospitals, Bicalutamide (Casadex) 150mg tab was the most expensive requiring 753 days wages, followed by Trastuzumab (Herceptin) 600mg vial which required 625 days wages, whilst Goserelin (Zoladex) 10.8mg vial required 124 days wages. The most affordable was Liposomal Doxorubicin (Caelyx) 50mg vial which required 2 days wages (See S8 Table).

In the private hospitals, Bevacizumab (Avastin) 400mg vial was the most expensive requiring 2554 days wages, followed by Trastuzumab (Herceptin) 600mg vial which required 753 days wages, whilst Abiraterone (Zytiga) 250mg tabs required 399 days wages. The most affordable was Anastrozole (Arimidex) 1mg tab which required 10 days wages (See S9 Table).

In the private pharmacies, Filgrastim (Neupogen) 300mcg vial was the most expensive requiring 1117 days wages, followed by Bevacizumab (Avastin) 400mg vial which required 594 days wages, whilst Trastuzumab (Herceptin) 600mg vial required 562 days wages. The most affordable was Granisetron (Kytril) 3 mg vial which required 7 days wages (See S10 Table).

In all sectors, all treatments with LPGs, costs more than 1 day's wage. In the public hospitals, Chlorambucil 2mg tab was the most expensive requiring 201 days wages, followed by Gemcitabine 1000mg vial which required 176 days wages, whilst Leucovorin Calcium 50mg vial required 144 days wages. The most affordable was Methotrexate 50mg vial which required 6 days wages (See S8 Table).

Table 4. Affordability of OB and LPG in all sectors.

| Affordability | Mean LPG | Mean OB |
|---|---|---|
| Private Hospital | 105.64 | 653.57 |
| Public Hospital | 54.81 | 188.00 |
| Private Pharmacy | 134.83 | 222.06 |
| All Facilities | 103.42 | 304.30 |

In the private hospitals, Sorafenib 200mg tab was the most expensive requiring 415 days wages, followed by Gemcitabine 1000mg vial which required 319 days wages, whilst Bleomycin 15 IU PFR vial required 179 days wages. The most affordable was Vincristine 1mg vial which required 11 days wages (See S9 Table).

In the private pharmacies, Thalidomide 50mg cap was the most expensive requiring 1642 days wages, followed by Imatinib 400mg tab which required 575 days wages, whilst Cytarabine 100mg vial required 527 days wages. The most affordable was Methotrexate 2.5mg tab which required 1.4 days wages (See S10 Table).

For the OBs with a comparator LPG, the cost of a one-month treatment with Bicalutamide 150mg tablets (OB) required about 753 days' wages and 8-days' wages for the LPG.

There were some instances where the OBs were more affordable than their comparator LPGs, e.g., for Cyclophosphamide 50mg tablets, people must work for 289 days to pay for the LPG, whereas they had to work for 43 days to pay for the OB.

## Discussion

In general, the availability of essential medicines has been problematic in sub-Saharan Africa, more than in any other region [46]. The average availability of essential medicines in sub-Saharan Africa is around 40% in the public sector and 60% in the private sector, substantially below the WHO target of 80% medicines availability in all sectors [47].

Our study showed very low availability of the cancer medicines in Ghana. It was disconcerting to see that none of the surveyed cancer medicines met the ideal availability WHO benchmark of 80% in all sectors [48]. The LPGs were more available than the OBs in all sectors and was more available in hospitals than in pharmacies. Conversely, the OBs were more available in the private facilities than the public facilities. Low availability of cancer medicines in public hospitals in most instances, forces the patient to access the private sector to buy expensive cancer medicines, resulting in financial hardships due to the higher out-of-pocket payments. High out-of-pocket costs for treatments are a key limitation to accessing health services. This highlights the need for government to streamline and improve cancer medicine procurement, distribution, and supply systems in the public system [7, 23, 49].

In Ghana, there are limited suppliers of cancer medicines due to the huge cost associated with its acquisition and supply. Most pharmacies do not routinely stock cancer medicines unless specifically requested for by prescribers, thus having unreliable availability even for those who can afford. There have been cases of stock outs of cancer medicines leading to frustration amongst patients coupled with loss of confidence in the health care system. Other reasons for the poor availability of cancer medicines such as, budgetary constraints, poor estimation of demands, delays in receiving the medication orders, in-country distribution challenges, lack of supplier motivation, and in accurate determination of needs, should be addressed [7, 50].

Our findings are consistent with studies in LMICs showing disparities in the actual availability and formulary availability of essential cancer medicines [7, 8, 23–25, 27, 29, 43, 51]. The WHO's EML for adults and EMLc for children is a guide to help countries develop their NEML and reimbursable list for the public sector, to improve the procurement, availability and use of adult and childhood cancers medicines respectively [34, 35]. While not a direct measure of availability, listing is an important step, to guide procurement and ensure availability of the cancer medicines [29]. The government should ensure that an up-to-date list of the Ghana EML and health insurance lists is maintained. At least for cytotoxic agents, there seems little value in maintaining separate NEMLs for adults and children as the same medicines are used in each group [8, 29].

Strong country pharmaceutical pricing policies can improve the affordability of pharmaceutical products when carefully planned, carried out, regularly checked, and revised according to changing conditions [45]. Differences in pricing policies account for the varying prices of medicines [24].

Cancer care in Ghana is hampered by several factors including high cost of cancer medicines and little reimbursement by insurance. Many medicine purchases by the hospitals are small in volume and therefore, high in acquisition and selling price. In Ghana, only cervical and breast cancer medicines are eligible for reimbursement under the NHIS [11]. The reimbursement of cervical and breast cancer medicines is based on the median generic price, thus forcing dispensers to dispense and patients to buy only generics of these cancer medicines. Patients can obtain cancer medicines only by prescription. Patients must pay for most cancer medicines out-of-pocket and face financial hardship due to unaffordable medical cost. This may lead to premature termination or interruption in treatment as seen also in other studies [26]. Ghana has a NMP to improve medicine pricing to ensure affordability [16]. This may not be strictly implemented as the government sets the price for the NHIS cancer medicines only. Importation of all cancer medicines is largely affected by forex fluctuations and determined by the pharmaceutical wholesalers/suppliers based on their production, marketing cost, taxes, duties, and desired benefit (30–40% for taxes and tariffs, and 50–200% for markups) [16].

The results of this study show that cancer medicine prices in Ghana are high, and there are large price differences amongst the sectors, between HPM and LPM, as well as between OB and LPG. The private sectors (private pharmacy and private hospitals) had the most expensive LPG and OB prices compared to the public sector (public hospitals). The private pharmacies had the lowest OB median prices, probably because of some pharmacies being suppliers/wholesalers of cancer medicines to both public and private pharmacies. Whilst the public hospitals had the lowest LPG median prices probably due to government policies on generic prescribing and medical insurance reimbursements. Several factors may explain the variability in prices paid by patients in the public and private sectors. These include, the sectors having different purchasing and distribution efficiency, the use of medicine sales as a cost-recovery mechanism to finance operating expenses and the unregulated and highly variable prices and mark-ups [32]. Overall, none of the medicines had a price ratio<1, suggesting inefficient procurement of cancer medicines in Ghana. The price variability between the OB and LPG, showed that when some OB medicines are prescribed/dispensed in Ghana, patients pay more than they would pay for generics. However, some LPGs were more expensive than the OBs showing efficient generic procurement and pricing probably due to generic competition, price-fixing, market competition or some other factors [23]. This is consistent with studies which showed that the existence of generics on the market might affect originator prices, or the originator prices remained at a high level [25]. The high patient prices could be due to OB patent protection, lack of generic competition, suppliers of generic medicines pricing popular products only slightly below the originator brand version, high manufacturer profit margins, high government taxes and duties on medicines, inefficient supply system and high wholesale or retail mark-ups [21].

Similar findings to our study, were also seen in studies conducted in LMICs (Countries; Nepal, India, Malaysia, and Regions; Africa, Latin America, Southeast Asia, Western Pacific, East Mediterranean) on pricing of cancer medicines [17–23, 41, 43, 51]. These published studies on the pricing of both adult and pediatric cancer medicines showed considerable variability in the prices of cancer medicines within a country, including wide variations in prices in the individual medicine categories and between brands. The price variation in public versus private facilities was also evident [18–23, 43].

Over 85% of OB/LPG medicines in the Public Hospitals (89.66%), Private Pharmacies (91.30%) and Private Hospitals (88.24%) were moderately inexpensive relative to the IRP, and may suggest effective procurement, or compromised quality control etc. [39]. This cannot be ascertained as there are some ambiguities when making international comparisons, due to varying medicine components, such as market size, market penetration pricing mechanisms, scale of economy, and taxation [41]. Further research to investigate reasons behind this variation is needed.

Government of Ghana (GoG) should strengthen cancer pricing policies to encompass equitable, affordable, and timely access, such that cancer patients have access to cancer medicines in a fair and timely manner without compromising the quality and safety of medicines. The patients should be able to afford cancer medicines over the full course of treatment. There should be good governance with appropriate prescribing, dispensing, and pricing. Procurement processes should include principles of transparency, efficiency, and accountability [5]. Government should launch initiatives to ensure patients have access to cheaper cancer medicines. They should promote quality assured generics and generic prescribing by physicians, improve pricing and price transparency, support cost sharing, encourage value-based reimbursement by medical insurance companies and grant faster approvals to new cancer medicines [5, 6, 19, 25, 43, 52–55]. They should also apply tax reduction or exemption for potential local manufacturing companies and companies that supply cheap cancer medicines and ensure savings are transferred to consumers [5]. Pricing policies and regulations such as tendering, negotiations, Single Exit Pricing (SEP), tiered pricing, value-based pricing, cost plus pricing, External Reference Pricing (ERP) or IRP and mark-up regulations across the supply and distribution chain should be explored and implemented to restrain medicine costs [20–22, 52].

Most patients in Ghana cannot afford the cancer medicines because of poverty, high inflation, and increasing cost of living. They sometimes present late for treatment, succumb to using herbal preparations and seek traditional spiritual remedies [30]. In developing countries like Ghana, where the medical insurance is in a developing stage, higher direct out of pocket payments costs means a decreased adherence to therapy, or default on treatment with its resulting disastrous implications and devastating impoverishing effects for families and households [56]. Affordability of cancer medicines in Ghana is a big challenge, as all treatments using standard regimens costs more than 1 day's wage. Cancer medicines are more affordable in the public sector (public hospitals) than in the private sectors (private pharmacies and private hospitals). The LPGW need 2554- days and 1642-days wages to afford a month's treatment with Bevacizumab 400mg vial (OB) and Thalidomide 50mg capsules (LPG), costing 5286.40 USD and 3399.82 USD respectively. This shows that a month's wage would be woefully inadequate to afford some cancer medicines and could pose problems for public health. It is important to bear in mind that these costs refer only to the medicine component of the total treatment costs. Consultation fees and diagnostic tests may mean that the total cost to the patient is considerably higher.

The results of our findings are consistent with studies showing patterns of affordability amongst countries, unaffordability of LPGs and OBs (adult and pediatric) in public and private facilities, and in some instances leading to treatment abandonment [23, 24, 27, 28, 43].

Africa including Ghana's health challenges are numerous and wide-ranging, with many factors affecting access to cancer medicines [7, 29, 57–59]. The GoG should implement policies that modify current high prices, fund cancer (adult and pediatric) medicines within the national budget, streamlines public and private sector procurement and supply systems to ensure equitable affordability in Ghana [21, 27].

## Limitations

This study, using basic indicators only, cannot give a complete picture of the pharmaceutical sector in Ghana. International price comparisons include many assumptions that affects its outcome, and thus are only pointers for further investigation. Some cancer medicines do not have IRPs, and this makes it difficult when computing the MPRs. The MPRs were calculated using the only available and outdated 2015 MSH IRPs, with prices deflated using the CPI to get a reliable MPR calculation. To resolve this issue, the HAI experts suggest avoiding calculating it and rely on MUPs only, thus we have presented the MUPs also. Results on affordability may also lead to over-estimation since the calculation used was based on the LPGWs wages. A significant proportion of the population especially patients with jobs in the informal sector earn less than the LPGW. The exclusion of the accompanying cost factors that play a role in the final cost to the cancer patient such as dispensing fees, facility fees, administration fees, doctors' fees and treatment courses requiring more than one medicine means that patients pay more. Other means of estimating affordability such as the catastrophic and impoverishment methods were not used. Use of retail prices, which include add-ons such as taxes and distribution fees was not estimated to identify potential targets for price reduction. Availability had a limitation of not capturing the pattern of medicine availability over time as it was measured at 'one time' on the day of data collection from the health facilities. Availability may be better understood through a longitudinal study instead of a cross-sectional study.

## Conclusion

Cancer medicine prices in Ghana are high, with large price differences. Availability was particularly very low for the cancer medicines and most patients cannot afford the cancer medicines. The cancer medicines should be available and affordable to reduce the cancer burden and mortality. The Government of Ghana must address determinants of health holistically with comprehensive policies and multifaceted interventions and regulations that modify current high cancer prices, provides tax incentives, health insurance and use of generics. This is to ensure equitable availability, timely affordable access, and appropriate use of quality cancer medicines irrespective of ability to pay at the point of use, to achieve healthy lives in line with the Ghana National Health Policy 2020.

## Supporting information

**S1 Table. Cancer medicines found in facilities, conditions and listing on the NEML, WHO EML, WHO EMLc.**
(PDF)

**S2 Table. Price variations of cancer medicine(s) in public hospitals.**
(PDF)

**S3 Table. Price variations of cancer medicine(s) in private hospitals.**
(PDF)

**S4 Table. Price variations of cancer medicine(s) in private pharmacies.**
(PDF)

**S5 Table. MPR of cancer medicines in public hospitals.**
(PDF)

**S6 Table. MPR of cancer medicines in private hospitals.**
(PDF)

**S7 Table. MPR of cancer medicines in private pharmacies.**
(PDF)

**S8 Table. Affordability of cancer medicines in public hospitals.**
(PDF)

**S9 Table. Affordability of cancer medicines in private hospitals.**
(PDF)

**S10 Table. Affordability of cancer medicines in private pharmacies.**
(PDF)

**S1 Data.**
(XLSX)

## Acknowledgments

We thank Divine Darlington Logo (HRU, GHS), for his support in data collection. We thank all personnel, institutions, pharmacies, and hospitals that provided access to their facilities during data collection. We thank Dr. Margaret Ewen (HAI/WHO), Dr Syed Hasan (University of Huddersfield), and Dr Amna Saeed (Lahore Medical and Dental College) for their support.

## Author Contributions

**Conceptualization:** Phyllis Ocran Mattila, Zaheer Ud-Din Babar.

**Data curation:** Phyllis Ocran Mattila.

**Formal analysis:** Phyllis Ocran Mattila.

**Funding acquisition:** Phyllis Ocran Mattila.

**Investigation:** Phyllis Ocran Mattila.

**Methodology:** Phyllis Ocran Mattila, Zaheer Ud-Din Babar.

**Project administration:** Phyllis Ocran Mattila.

**Resources:** Phyllis Ocran Mattila.

**Supervision:** Richard Berko Biritwum, Zaheer Ud-Din Babar.

**Validation:** Phyllis Ocran Mattila.

**Writing – original draft:** Phyllis Ocran Mattila.

**Writing – review & editing:** Phyllis Ocran Mattila, Zaheer Ud-Din Babar.

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
