## [Decision Letter · Decision Letter 0]

12 Jul 2022

PONE-D-21-38310A Comprehensive Survey of Cancer Medicines Pricing, Availability and Affordability in GhanaPLOS ONE

Dear Dr. Ocran Mattila,

Thank you for submitting your manuscript to PLOS ONE. After careful consideration, we feel that it has merit but does not fully meet PLOS ONE’s publication criteria as it currently stands. Therefore, we invite you to submit a revised version of the manuscript that addresses the points raised during the review process.

The manuscript has been evaluated by four reviewers, and their comments are available below.

The reviewers have raised a number of major concerns. They feel the manuscript should outline a clearly-defined research question, and they request improvements to the reporting of methodological aspects of the study. The reviewers also note concerns about the presentation of the results and data analysis.

Could you please carefully revise the manuscript to address all comments raised?

We look forward to receiving your revised manuscript.

Kind regards,

Thomas Phillips, PhD

Staff Editor

PLOS ONE

Journal Requirements:

2. Our staff editors have determined that your manuscript is likely within the scope of our Cancer and Social Inequity Call for Papers. This editorial initiative is headed by a team of Guest Editors for PLOS ONE: Vesna Zadnik (Institute of Oncology, Ljubljana), Nixon Niyonzima (Uganda Cancer Institute), Claudia Allemani (London School of Hygiene and Tropical Medicine). This call for papers aims to highlight the negative impacts of social inequities on health, identify the effects of social and corporate policies on access to healthcare services, and propose solutions to promote more equitable cancer outcomes and ultimately, social justice.  Additional information can be found on our announcement page: https://collections.plos.org/call-for-papers/cancer-and-social-inequality/

If you would like your manuscript to be considered for this collection, please let us know in your cover letter and we will ensure that your paper is treated as if you were responding to this call.  Please note that being considered for the Collection does not require additional peer review beyond the journal’s standard process and will not delay the publication of your manuscript if it is accepted by PLOS ONE. If you would prefer to remove your manuscript from collection consideration, please specify this in the cover letter.

Reviewers' comments:

Reviewer's Responses to Questions

**Comments to the Author**

1. Is the manuscript technically sound, and do the data support the conclusions?

Reviewer #1: Yes

Reviewer #2: Yes

Reviewer #3: Yes

Reviewer #4: Yes

2. Has the statistical analysis been performed appropriately and rigorously? 

Reviewer #1: Yes

Reviewer #2: Yes

Reviewer #3: I Don't Know

Reviewer #4: Yes

3. Have the authors made all data underlying the findings in their manuscript fully available?

Reviewer #1: Yes

Reviewer #2: Yes

Reviewer #3: No

Reviewer #4: Yes

4. Is the manuscript presented in an intelligible fashion and written in standard English?

Reviewer #1: Yes

Reviewer #2: Yes

Reviewer #3: Yes

Reviewer #4: No

5. Review Comments to the Author

Reviewer #1: The study presents relevant data and findings on availability, prices, and affordability of cancer medicines in Ghana. This study uses an adapted version of a standard methodology developed and widely used by Health Action International (HAI) and WHO. It is a relevant topic and study on the discussion of access to cancer medicines in Low- and Middle-Income countries. This is one of the few studies that has used the HAI methodology to measure access to cancer medicines. The reviewer commends and congratulates the authors on doing such considerable work on the adaptation of the methodology and undertaking the task of doing the survey to measure access to medicines in Ghana.

However, the reviewer has comments that need addressing

1. Abstract - the reviewer feels that the abstract is incomplete and would benefit of a sentence to summarize the main conclusion(s) of the manuscript.

2. Introduction - the reviewer recommends the authors to revise the whole introduction. I feel that the flow of information is not consistent nor tells a story. The second paragraph of the introduction feels disconnected from the previous one. In addition, the first sentece of this paragraph is not relates to the following sentences nor it is relevant. I suggest the authors to explain clearly how the health system works in Ghana in terms of access to medicines, for cancer specifically, so that the reader can clearly understand the context. Please revise all paragraphs and make sure that they follow a consistent flow of ideas and concepts that are relevant for the study and to set the scene.

3. Methods - the reviewer recommends to the authors to explain (as mentioned in the previous comment) how is Ghana set in terms of access to medicines to better explain the rationale on the selection of outlets and facilities that were surveyed. Furthermore, in Table 2, all surveyed medicines are listed and include information of whether they are listed or not in the Ghanean EML, WHO EML and Children's EML, and there are some medicines that are not included in any EML. The selection criteria of the medicines of study needs to be better explained and justified, since the HAI suggests and promotes to survey essential medicines.

I suggest to the authors to better explain the considerations taken on how affordability was measured (suggest to revise: https://doi.org/10.1186/s12913-020-05167-9). Since not all cancer medicines are tablets taken daily, and most treatments for cancer take more than one medicine in different doses according to a person's size, one would need to consider that some treatments do not take one month, but one take, and that a patient would not use a whole vial (for intravenous medicines) of one medicine for their treatment. The formula use and the considerations on how affordability was calculated need to be better explained and justified, so that other countries that might want to replicate the study, can understand and replicate the methods.

4. Results - there are some sentences in the results sections that can be better places in the discussion section, as they actually discuss or analyze the implications of the results. In the results section, it is not clear for me (nor in the text nor in the titles of the tables) where are the international comparisons (use of the IRP). Please clarify and make sure it is easy for the ready to find these results.

In the results and the discussion, please keep the same order as in the methods sections. That will make the manuscript more reader friendly and will keep a consistent order of ideas, findings, and analyses and reflections on the results.

5. Discussion - as mentioned above, please keep the same order of ideas and concepts as in the results section. I feel that the authors need to revise their reflections on their results and analyze what does that mean for access to medicines in Ghana, and include recommendations for action for the government. But this needs to be done in a more structured manner. I suggest to keep their reflections of their findings within the concepts studies (availability, prices, and affordability) and refrain to go into further topics that influence and affect access to medicines (e.g. faster approvals of medicines, etc.). In this section, the authors can also give more context and more explanations on their recommendations and why would these improve the situation.

In the discusison of prices of medicines, the authors have included a comment that suggest the empowerment of patients to shop around for cheaper medicines. I suggest to the authors to revise this comment. Medicines should be available and affordable in medicines outlets so that patients can receive treatment at the moment needed. If this is a common practice in Ghana, I would suggest to give more context and explain why this is a recommendation.

I would suggest to the authors to move the 'Limitations' section to the discussion. I would suggest the authors to structure their discussion starting with reflections and actual discussion on their findings, followed by an implications section where policy options can be discussed and recommended, and then include the limitations section.

6. Conclusion - the conclusion feels a bit disconnected from the discussion, as it introduces new concepts that should rather be in the discussion. The conclusions should give a summary of the main findings with their implications on access to medicines and needed actions.

The authors need to revise their sentences and grammar, since there are some sentences that seem incomplete, and others and not very clear. The reviewer has provided more detailed comments on the file attached.

Reviewer #2: Introduction

The authors mentioned that there are 29 cancer facilities in Ghana. The authors should check and correct that statement because the cancer centers are all specialist referral centers, primarily public, a few private centers. The other hospitals are referral hospitals, and they are not centers for the diagnosis and treatment of cancer.

The authors stated that one of the reasons why most cancer patients do not have affordable access to treatment is the absence of cancer medicines on the EML.In Ghana, cancer medicines are included in the EML. The current version was published in 2017 and its available online (moh.gov.gh/wp-content/uploads/2020/07/Ghana-EML-2017.pdf).

They claim that no study has been conducted on the availability, price, and affordability of medicines in Ghana. However, a similar study was done in Ghana and published in BMC cancer recently (Evaluating essential medicines for treating childhood cancers: availability, price and affordability study in Ghana). The authors should amend their statements.

Method

The major components (price, availability and affordability) should be analyzed under different sub-headings instead of lumping all of them together as one. This gives a better understanding and also helps readers to appreciate the study.

The authors stated that the data collection sheet or form is available upon request but failed to describe how the data was collected from the survey sites. Kindly address it.

The authors stated that the collection of cancer medicines was based on chemotherapy of neoplastic diseases in Ghana. They should support this statement with reference(s). They should also state the year for the WHO EML, NEML and EMLc. For example, 2017 WHO EML.

The authors assessed the affordability of cancer medicines based on standard regimens. The authors should provide a table showing the cost of the regimen and indicating the number of daily wages to be able to afford these medications using the lowest prices. Also, the authors can use the most prevalent cancer type in Ghana as an example to calculate the affordability.

The authors use Bivacizumab as an example to calculate the affordability which is not commonly used in most of the regimens in Ghana. The standard treatment guideline of Ghana provides the typical regimen, which can serve as an example to calculate for the affordability of medicines other than the less commonly used biological agent.

Reviewer #3: The topic is interesting and seems to be fully investigated. But, the methods are not clear and results are nor well presented. Please, if there is no political and/or ethical issues, send also the data collection tool generated from the HAI excel worksheet (in case you have really used the WHO/HAI methods) and the worksheet saved after data entry.

Comments were grouped in major and minor comments.

A. Major comments

1. Format:

a. The text lines should be numbered to ease the review and reference of the comments. Please, do consider every note to authors. To insert lines numbers, go to “Layout” and select continuous lines numbers. Avoid page breaks to keep lines numbers continuous.

2. The abstract should be organized in background, methods, results, and conclusion.

3. Introduction:

a. Match the statement with the content.

b. When you write “Several studies have examined the availability, price, and affordability of medicines” we expect you to provide at least three references, but you gave only one reference. Please, rephrase the statement or, add more references.

c. This statement sounds incorrect: “This study, which to the best of our knowledge, is the first national comprehensive survey to be conducted in Ghana.”

d. Looking at the methodology and results, you only covered “Public Hospitals, Private Hospitals, and Private Pharmacies of Ghana” and “A total of 29 facilities (7 public hospitals, 20 private pharmacies and 2 private hospitals) in 4 survey areas (Greater Accra, Ashanti, Central and Northern Regions respectively)”. Please, rephrase the text to reflect the coverage of your study. Unless these health facilities are the only ones providing anticancer treatment in Ghana.

4. Methods:

a. Wrong use of the defined methodology: presentation of more than one LPG, while the method published by WHO/HAI recommends selecting the generic brand with the lowest price (tables 5-7).

b. This statement/sentence is too long and confusing “For each cancer medicine, data was collected on its availability, and on the low, high and median patient retail unit price (including if only one price was found) of the OB, which is the original pharmaceutical product that was first authorized for marketing and the International Nonproprietary Name (INN)(generic name)/LPG, which is the lowest price found at each medicine outlet (place where medicines are dispensed to patients), of products other than the OB that contains the same active ingredient and marketed under a generic name”.

c. This statement is confusing and contraries the method mentioned of WHO/HAI “The price, availability, and affordability were presented as means and standard deviations.”

d. More confusion with the following statement “The differences in prices, availability and affordability were calculated using Kruskal-Wallis test and p< 0.05 was used to indicate a significant difference.” Please, describe the methods used during this survey. The WHO/HAI describes clearly how to calculate/measure prices, availability, and affordability, which is different from what you reported here (cfr page 7-8). Write and clearly describe this section to help readers understand the methods that you have used.

e. I failed to understand this text, it is confusing for me and I don’t catch what and how you actually did “The 2015 Management Sciences Health (MSH), Supplier IRP were used to compare local cancer medicine prices to an international standard [23]. The MPR is the ratio of median local unit price of a surveyed medicine to its (IRP), and was calculated as follows: MPR = Median local unit price (USD)/Median IRP (USD). For this study, the MPR was calculated using prices having a deflation factor of 84.73% according to the Ghana Consumer Price Indices (CPI) for July 2014 to July 2019 [24], as the latest available IRPs were published in 2015 [23]. Like other studies, this price adjustment method was adopted to make the comparison between local and international prices more reliable [25, 26].”

5. Results:

a. They should be presented in summary tables or figures and not in raw-data tables. Table 2 does not present results, rather data. But also, for each INN, you should select the strength to survey, find the Originator brand, and the Lowest Priced generic: just two rows for each INN, and you will have less than 65. Ideally, avoid tables taking more than an A4 page.

b. Table 3: please, carefully read the WHO/HAI methods to understand what kind of results you should report under “Availability”. The reported results are wrong and should be corrected. Please, report on availability of cancer medicines.

c. Figs 1a and 1b “Percentage Availability of LPG and OB using Kruskal Wallis Test” should be presented before putting their captions and cited in the text.

d. Figs 3a and 3b must be introduced by a text in which reference is made to them, instead of immediately follow table 4.

e. Fig 4: inappropriate presentation. What is the purpose of putting the median price and the median prices variation on the same scheme, to totalize 100%?

f. Wrong table titles: the ontent of tables is different from the table titles: e.g. tables 5-7;

g. Table 7: how do you have only 3 anticancer medicines instead of 65? Same question for the other tables where you have less than 65 anticancer medicines.

h. Wrong formulae to calculate the price ratio in table 5 (please, read carefully the methodology published by WHO/HAI) and keep your text consistent. You are mixing different methods and I failed to get the one you have used.

i. Wrong measurement/ presentation of affordability: affordability should be calculated and presented for medicines, not for categories of health facilities (Table 8).

j. Missing labels on figure: 6,

k. Duplicated results: Figure 7a (.docx and .tiff).

B. Minor comments

1. Title: Replace “Pricing” by “Prices” because pricing implies price components, and you have only measured prices.

2. Legend for authorships:

a. Write in full words the abbreviations (authors 1&3);

b. Delete the details about the courses taught by authors, limit the legend at their affiliation department, institution, and country.

c. For the corresponding author, do not repeat the legend presented above, just present the names and the e-mail address.

d. Harmonize the affiliations for authors: either University, School, Department or; Department, School then University. Corresponding author: just put the e-mail address, the other details are not needed.

3. Abstract:

a. Organize it in background, methods, results, and conclusion.

b. No need to abbreviate the country’s name, write it in full since this abbreviation has not been used elsewhere in your text;

c. What do you mean by cancer prices? Please, rephrase this statement in case you wanted to mean prices for cancer medicines.

d. Rephrase the statement “An adapted World Health Organization (WHO)/ Health Action International (HAI) methodology was used for measuring medicine prices.” To “the methods developed and standardized by the World Health Organization (WHO) in collaboration with the Health Action International (HAI) in (year of publication) was adapted and used to measure prices, availability, and affordability of cancer medicines in Ghana”.

e. Limit or avoid the use of abbreviations in the abstract.

f. Too much details “The availability of cancer medicines was assessed as percentage of health facilities stocked with listed medicines. The Price U.S dollars (USD) variation between the same medicine’s highest- and lowest-priced product, Originator Brand (OB) and its Lowest Priced Generic (LPG) was compared”.

g. Please, make your statements clear: the WHO/HAI methods measure prices in terms of Median Price Ratio (MPR); why did you change it to USD? Why do you mix USDs and MPRs?

h. The statement “The median prices (USD) of LPG ranged as (0.25 – 227.98), and OB (0.41-1321.60)” is grammatically incorrect and meaningless.

i. What do you mean by “Adjusted MPRs” and what kind of adjustment did you make?

j. Avoid mixing MPRs for LPGs and OBs: it is like counting together hens and eggs.

4. Introduction: Use the appropriate tense for reported cancer burden.

5. Mthods

a. Table 1 should be presented under the results section;

b. Inappropriate tense: e.g. “The availability of the cancer medicines at the specified strengths for both OB and LPG will be analyzed by the percentage (%) …..”

6. Results

a. Reference made to table 2, why did you include two different dosage strengths for the same brand of Bicalutamide? Carboplatine, …?

b. Figure 4, 7a, 7b, should be introduced and reference made to it before inserting its caption.

c. Figures 8, 9, 10 should wear the same number with a, b, and c.

Reviewer #4: 1.Is it reasonable to use LPGWs as the threshold for the affordability evaluation of cancer patients? Are there other evaluation criteria in similar studies？

2.It is recommended to mark the IRP in the included drug list.

3.What is the purpose of adding indications to the list of included drugs? There is not much value in the article.

4.Most antitumor drugs do not have IRP values. The authors should further explain when comparing MPR values. Although they are mentioned in the “Limitations” , the problems encountered in the actual calculation are not presented.

5.It is suggested that urban residents and rural residents should be discussed separately in the evaluation of affordability, showing a greater difference.

6.The patient's out-of-pocket ratio is not mentioned in the article.

7.Price is the biggest factor affecting the affordability and availability of cancer patients. The article should focus on the intervention and unification of price. For example, it is suggested that the government should control the price of anti-tumor. Drugs with the same specifications and dosage forms should have the same price, and private pharmacies or hospitals may give appropriate mark-ups on the basis of the unified price of public hospitals.

8.The discussion part of the article seems to talk less about the enterprise factors that cause high prices, and it is recommended to include them appropriately.

9.Figure 5 and Figure 6 are the presentation of the MPR . It may be clearer whether it can be made into two tables.

10.It is recommended that affordability should be evaluated by the indicator "catastrophic medical expenditures", In other words,WHO/HAI and "catastrophic medical expenditures" are applied to judge affordability, and to discuss the proportion and number of drugs from catastrophic medical expenditure.

6. PLOS authors have the option to publish the peer review history of their article (what does this mean?). If published, this will include your full peer review and any attached files.

Reviewer #1: No

Reviewer #2: **Yes: **Kofi Boamah Mensah

Reviewer #3: **Yes: **Thomas Bizimana

Reviewer #4: No

---

## [Author Response · Author response to Decision Letter 0]

24 Aug 2022

Reviewer 1- Ques 1-6 has been addressed. ( Ref. attached response to Reviewers comments).

Reviewer 2- All comments have been addressed. ( Ref. attached response to Reviewers comments).

Reviewer 3- All comments have been addressed. ( Ref. attached response to Reviewers comments).

Reviewer 4- All comments have been addressed. ( Ref. attached response to Reviewers comments).

Response to reviewers’ comments

1st Reviewer’s comment Authors’ response

Ques 1: 

Abstract - the reviewer feels that the abstract is incomplete and would benefit of a sentence to summarize the main conclusion(s) of the manuscript.

 Thank you for your comment. We have revised the text based on your feedback.

Ques 2:

Introduction - the reviewer recommends the authors to revise the whole introduction. I feel that the flow of information is not consistent nor tells a story. The second paragraph of the introduction feels disconnected from the previous one. In addition, the first sentence of this paragraph is not relates to the following sentences nor it is relevant. I suggest the authors to explain clearly how the health system works in Ghana in terms of access to medicines, for cancer specifically, so that the reader can clearly understand the context. Please revise all paragraphs and make sure that they follow a consistent flow of ideas and concepts that are relevant for the study and to set the scene. 

Thank you for your comment. We have revised the introduction text based on your feedback.

Ques 3:

Methods - the reviewer recommends to the authors to explain (as mentioned in the previous comment) how is Ghana set in terms of access to medicines to better explain the rationale on the selection of outlets and facilities that were surveyed. Furthermore, in Table 2, all surveyed medicines are listed and include information of whether they are listed or not in the Ghanean EML, WHO EML and Children's EML, and there are some medicines that are not included in any EML. The selection criteria of the medicines of study needs to be better explained and justified, since the HAI suggests and promotes to survey essential medicines.

I suggest to the authors to better explain the considerations taken on how affordability was measured (suggest to revise: https://doi.org/10.1186/s12913-020-05167-9). Since not all cancer medicines are tablets taken daily, and most treatments for cancer take more than one medicine in different doses according to a person's size, one would need to consider that some treatments do not take one month, but one take, and that a patient would not use a whole vial (for intravenous medicines) of one medicine for their treatment. The formula use and the considerations on how affordability was calculated need to be better explained and justified, so that other countries that might want to replicate the study, can understand and replicate the methods. 

Many thanks for the comments. The facilities were selected based on the availability of cancer medicines in that facility during the survey. So basically, every facility that provides cancer medicines in Ghana was included in the survey.

As this was a comprehensive national survey and noting that the Ghana EML was last updated in 2017, every cancer medicine used for cancer treatment in Ghana based on the local disease burden was included in the study.

For affordability calculations, as per the WHO/HAI method, A month of cancer medicines was used to demonstrate the economic implication on a patient if they would have to pay for it out of pocket, even though a cancer patient is expected to have more than one cycle of treatment with multiple medicine regimens. 

The cost of a course of therapy for important conditions can be compared with the daily wage of the lowest-paid unskilled government worker. This analysis is very valuable as an advocacy tool since it expresses prices in relation to an individual’s ability to pay rather than to international prices.

Ques 4: 

Results - there are some sentences in the results sections that can be better places in the discussion section, as they actually discuss or analyze the implications of the results. In the results section, it is not clear for me (nor in the text nor in the titles of the tables) where are the international comparisons (use of the IRP). Please clarify and make sure it is easy for the ready to find these results.

In the results and the discussion, please keep the same order as in the methods sections. That will make the manuscript more reader friendly and will keep a consistent order of ideas, findings, and analyses and reflections on the results.

Many thanks for the comments. It has been revised.

Ques 5:

Discussion - as mentioned above, please keep the same order of ideas and concepts as in the results section. I feel that the authors need to revise their reflections on their results and analyze what does that mean for access to medicines in Ghana, and include recommendations for action for the government. But this needs to be done in a more structured manner. I suggest to keep their reflections of their findings within the concepts studies (availability, prices, and affordability) and refrain to go into further topics that influence and affect access to medicines (e.g. faster approvals of medicines, etc.). In this section, the authors can also give more context and more explanations on their recommendations and why would these improve the situation.

In the discusison of prices of medicines, the authors have included a comment that suggest the empowerment of patients to shop around for cheaper medicines. I suggest to the authors to revise this comment. Medicines should be available and affordable in medicines outlets so that patients can receive treatment at the moment needed. If this is a common practice in Ghana, I would suggest to give more context and explain why this is a recommendation.

I would suggest to the authors to move the 'Limitations' section to the discussion. I would suggest the authors to structure their discussion starting with reflections and actual discussion on their findings, followed by an implications section where policy options can be discussed and recommended, and then include the limitations section. 

Many thanks for your comments. It has been revised.

Ques 6:

Conclusion - the conclusion feels a bit disconnected from the discussion, as it introduces new concepts that should rather be in the discussion. The conclusions should give a summary of the main findings with their implications on access to medicines and needed actions.

The authors need to revise their sentences and grammar, since there are some sentences that seem incomplete, and others and not very clear. The reviewer has provided more detailed comments on the file attached.

Many thanks for your comments. It has been revised.

2nd Reviewer’s Comments 

Ques 1:

Introduction

The authors mentioned that there are 29 cancer facilities in Ghana. The authors should check and correct that statement because the cancer centers are all specialist referral centers, primarily public, a few private centers. The other hospitals are referral hospitals, and they are not centers for the diagnosis and treatment of cancer.

The authors stated that one of the reasons why most cancer patients do not have affordable access to treatment is the absence of cancer medicines on the EML.In Ghana, cancer medicines are included in the EML. The current version was published in 2017 and its available online (moh.gov.gh/wp-content/uploads/2020/07/Ghana-EML-2017.pdf).

They claim that no study has been conducted on the availability, price, and affordability of medicines in Ghana. However, a similar study was done in Ghana and published in BMC cancer recently (Evaluating essential medicines for treating childhood cancers: availability, price and affordability study in Ghana). The authors should amend their statements.

Thank you for your comment. We have revised the introduction text based on your feedback. 

The 29 cancer facilities (public hospitals, private hospitals and private pharmacies are those providing cancer medicines to patients. 

The current version of the Ghana EML, 2017 does not list all the cancer medicines found during the survey. (Ref Table 2). While not a direct measure of availability, listing is an important step, to guide procurement and ensure availability of the cancer medicines.

The claim that no study has been conducted has been modified to reflect the first national comprehensive survey of adults’ cancer medicines pricing, availability, and affordability in Ghana’, as what was published in BMC was on childhood cancers.

Ques 2:

Method

The major components (price, availability and affordability) should be analyzed under different sub-headings instead of lumping all of them together as one. This gives a better understanding and also helps readers to appreciate the study.

The authors stated that the data collection sheet or form is available upon request but failed to describe how the data was collected from the survey sites. Kindly address it.

The authors stated that the collection of cancer medicines was based on chemotherapy of neoplastic diseases in Ghana. They should support this statement with reference(s). They should also state the year for the WHO EML, NEML and EMLc. For example, 2017 WHO EML.

The authors assessed the affordability of cancer medicines based on standard regimens. The authors should provide a table showing the cost of the regimen and indicating the number of daily wages to be able to afford these medications using the lowest prices. Also, the authors can use the most prevalent cancer type in Ghana as an example to calculate the affordability.

The authors use Bivacizumab as an example to calculate the affordability which is not commonly used in most of the regimens in Ghana. The standard treatment guideline of Ghana provides the typical regimen, which can serve as an example to calculate for the affordability of medicines other than the less commonly used biological agent. 

Many thanks for the comments. They have been addressed. Table on cost of regimen has been included, which shows all other medicines in addition to bevacizumab.

Some percentage of patients are on Bivacizumab, thus it was used to show affordability for such group of patients. The table shows all the medicines. 

3rd Reviewer’s Comments 

Ques 1:

The topic is interesting and seems to be fully investigated. But, the methods are not clear and results are nor well presented. Please, if there is no political and/or ethical issues, send also the data collection tool generated from the HAI excel worksheet (in case you have really used the WHO/HAI methods) and the worksheet saved after data entry.

Comments were grouped in major and minor comments.

A. Major comments

1. Format:

a. The text lines should be numbered to ease the review and reference of the comments. Please, do consider every note to authors. To insert lines numbers, go to “Layout” and select continuous lines numbers. Avoid page breaks to keep lines numbers continuous.

Thank you for your comment. We have revised the text based on your feedback. Text lines have been numbered. Please send me an email for the data collection tool.

Ques 2:

The abstract should be organized in background, methods, results, and conclusion.

Thank you for your comment. We have revised the abstract based on your feedback.

Ques 3:

Introduction:

a. Match the statement with the content.

b. When you write “Several studies have examined the availability, price, and affordability of medicines” we expect you to provide at least three references, but you gave only one reference. Please, rephrase the statement or, add more references.

c. This statement sounds incorrect: “This study, which to the best of our knowledge, is the first national comprehensive survey to be conducted in Ghana.”

d. Looking at the methodology and results, you only covered “Public Hospitals, Private Hospitals, and Private Pharmacies of Ghana” and “A total of 29 facilities (7 public hospitals, 20 private pharmacies and 2 private hospitals) in 4 survey areas (Greater Accra, Ashanti, Central and Northern Regions respectively)”. Please, rephrase the text to reflect the coverage of your study. Unless these health facilities are the only ones providing anticancer treatment in Ghana.

Thank you for your comment. 

a. We have revised the text based on your feedback. 

b. More references have been included.

c. This has been modified to reflect the first national comprehensive survey of adults’ cancer medicines pricing, availability, and affordability in Ghana’, as what was published in BMC was on childhood cancers.

d. these facilities are the only ones providing cancer medicines in Ghana

Ques 4:

Methods:

a. Wrong use of the defined methodology: presentation of more than one LPG, while the method published by WHO/HAI recommends selecting the generic brand with the lowest price (tables 5-7).

b. This statement/sentence is too long and confusing “For each cancer medicine, data was collected on its availability, and on the low, high and median patient retail unit price (including if only one price was found) of the OB, which is the original pharmaceutical product that was first authorized for marketing and the International Nonproprietary Name (INN)(generic name)/LPG, which is the lowest price found at each medicine outlet (place where medicines are dispensed to patients), of products other than the OB that contains the same active ingredient and marketed under a generic name”.

c. This statement is confusing and contraries the method mentioned of WHO/HAI “The price, availability, and affordability were presented as means and standard deviations.”

d. More confusion with the following statement “The differences in prices, availability and affordability were calculated using Kruskal-Wallis test and p< 0.05 was used to indicate a significant difference.” Please, describe the methods used during this survey. The WHO/HAI describes clearly how to calculate/measure prices, availability, and affordability, which is different from what you reported here (cfr page 7-8). Write and clearly describe this section to help readers understand the methods that you have used.

e. I failed to understand this text, it is confusing for me and I don’t catch what and how you actually did “The 2015 Management Sciences Health (MSH), Supplier IRP were used to compare local cancer medicine prices to an international standard [23]. The MPR is the ratio of median local unit price of a surveyed medicine to its (IRP), and was calculated as follows: MPR = Median local unit price (USD)/Median IRP (USD). For this study, the MPR was calculated using prices having a deflation factor of 84.73% according to the Ghana Consumer Price Indices (CPI) for July 2014 to July 2019 [24], as the latest available IRPs were published in 2015 [23]. Like other studies, this price adjustment method was adopted to make the comparison between local and international prices more reliable [25, 26].”

Many thanks for the comments. We have revised this section. 

a. Revised

b. Revised 

c. revised

d. This refers to statistical analysis

e. Method has been revised to explain better. Please see reference 26a and 26 b to understand the deflation/inflation factor used as the MSH prices are old and the calculations needed to reflect 2020 prices.

Ques 5:

Results:

a. They should be presented in summary tables or figures and not in raw-data tables. Table 2 does not present results, rather data. But also, for each INN, you should select the strength to survey, find the Originator brand, and the Lowest Priced generic: just two rows for each INN, and you will have less than 65. Ideally, avoid tables taking more than an A4 page.

b. Table 3: please, carefully read the WHO/HAI methods to understand what kind of results you should report under “Availability”. The reported results are wrong and should be corrected. Please, report on availability of cancer medicines.

c. Figs 1a and 1b “Percentage Availability of LPG and OB using Kruskal Wallis Test” should be presented before putting their captions and cited in the text.

d. Figs 3a and 3b must be introduced by a text in which reference is made to them, instead of immediately follow table 4.

e. Fig 4: inappropriate presentation. What is the purpose of putting the median price and the median prices variation on the same scheme, to totalize 100%?

f. Wrong table titles: the ontent of tables is different from the table titles: e.g. tables 5-7;

g. Table 7: how do you have only 3 anticancer medicines instead of 65? Same question for the other tables where you have less than 65 anticancer medicines.

h. Wrong formulae to calculate the price ratio in table 5 (please, read carefully the methodology published by WHO/HAI) and keep your text consistent. You are mixing different methods and I failed to get the one you have used.

i. Wrong measurement/ presentation of affordability: affordability should be calculated and presented for medicines, not for categories of health facilities (Table 8).

j. Missing labels on figure: 6,

k. Duplicated results: Figure 7a (.docx and .tiff).

 Many thanks for your comments.

a. Table 2 is like what is in Ref 26b. the 65 refers to individual medicines (OB, INN/LPG).

b. Table 3 shows the statistical report of the mean of median (average of all MUPs) availability of the LPGs and OBs in all public hospitals, all private hospitals, and all private Pharmacies. 

c. Done

d. done

e. revised

f. revised. 

g. Table 7 and all the other tables 5 and 6 didn’t show the full 65 medicines as the analysis was based on matched pairs. Thus, the same medicines having different prices in the facilities such that the highest price and the lowest price could be obtained was used for the analysis.

h. Price ratio was calculated as; Price of the OB divided by Price of the LPG or Price of the HPM divided by Price of the LPM. Please note, this is a calculation of price ratios and not median price ratios.

i. Table on affordability included to show calculations.

j. Done

k. Done

Ques 6:

B. Minor comments

1. Title: Replace “Pricing” by “Prices” because pricing implies price components, and you have only measured prices.

2. Legend for authorships:

a. Write in full words the abbreviations (authors 1&3);

b. Delete the details about the courses taught by authors, limit the legend at their affiliation department, institution, and country.

c. For the corresponding author, do not repeat the legend presented above, just present the names and the e-mail address.

d. Harmonize the affiliations for authors: either University, School, Department or; Department, School then University. Corresponding author: just put the e-mail address, the other details are not needed.

3. Abstract:

a. Organize it in background, methods, results, and conclusion.

b. No need to abbreviate the country’s name, write it in full since this abbreviation has not been used elsewhere in your text;

c. What do you mean by cancer prices? Please, rephrase this statement in case you wanted to mean prices for cancer medicines.

d. Rephrase the statement “An adapted World Health Organization (WHO)/ Health Action International (HAI) methodology was used for measuring medicine prices.” To “the methods developed and standardized by the World Health Organization (WHO) in collaboration with the Health Action International (HAI) in (year of publication) was adapted and used to measure prices, availability, and affordability of cancer medicines in Ghana”.

e. Limit or avoid the use of abbreviations in the abstract.

f. Too much details “The availability of cancer medicines was assessed as percentage of health facilities stocked with listed medicines. The Price U.S dollars (USD) variation between the same medicine’s highest- and lowest-priced product, Originator Brand (OB) and its Lowest Priced Generic (LPG) was compared”.

g. Please, make your statements clear: the WHO/HAI methods measure prices in terms of Median Price Ratio (MPR); why did you change it to USD? Why do you mix USDs and MPRs?

h. The statement “The median prices (USD) of LPG ranged as (0.25 – 227.98), and OB (0.41-1321.60)” is grammatically incorrect and meaningless.

i. What do you mean by “Adjusted MPRs” and what kind of adjustment did you make?

j. Avoid mixing MPRs for LPGs and OBs: it is like counting together hens and eggs.

4. Introduction: Use the appropriate tense for reported cancer burden.

5. Mthods

a. Table 1 should be presented under the results section;

b. Inappropriate tense: e.g. “The availability of the cancer medicines at the specified strengths for both OB and LPG will be analyzed by the percentage (%) …..”

6. Results

a. Reference made to table 2, why did you include two different dosage strengths for the same brand of Bicalutamide? Carboplatine, …?

b. Figure 4, 7a, 7b, should be introduced and reference made to it before inserting its caption.

c. Figures 8, 9, 10 should wear the same number with a, b, and c.

Many thanks for your comments.

1. It has been revised accordingly.

2. a-done

b-done

c-done

d-done

3. a-done

b-done

c-done

d-rephrased

e-limited use of abbreviations 

f- reworded

g- reworded

h- reworded

i-to adjust the MPRs as the MSH prices for comparison was in 2015, and hence the need to deflate/inflate them to 2020 prices to enable realistic comparisons.

J-done

4. done

5. a. done. 

b. revised

6. a. Each dosage strength was taken as a separate medicine, thus included for analysis.

b. done

c. done

Reviewer #4: 1.Is it reasonable to use LPGWs as the threshold for the affordability evaluation of cancer patients? Are there other evaluation criteria in similar studies？

2.It is recommended to mark the IRP in the included drug list.

3.What is the purpose of adding indications to the list of included drugs? There is not much value in the article.

4.Most antitumor drugs do not have IRP values. The authors should further explain when comparing MPR values. Although they are mentioned in the “Limitations” , the problems encountered in the actual calculation are not presented.

5.It is suggested that urban residents and rural residents should be discussed separately in the evaluation of affordability, showing a greater difference.

6.The patient's out-of-pocket ratio is not mentioned in the article.

7.Price is the biggest factor affecting the affordability and availability of cancer patients. The article should focus on the intervention and unification of price. For example, it is suggested that the government should control the price of anti-tumor. Drugs with the same specifications and dosage forms should have the same price, and private pharmacies or hospitals may give appropriate mark-ups on the basis of the unified price of public hospitals.

8. The discussion part of the article seems to talk less about the enterprise factors that cause high prices, and it is recommended to include them appropriately.

9.Figure 5 and Figure 6 are the presentation of the MPR . It may be clearer whether it can be made into two tables.

10.It is recommended that affordability should be evaluated by the indicator "catastrophic medical expenditures", In other words, WHO/HAI and "catastrophic medical expenditures" are applied to judge affordability, and to discuss the proportion and number of drugs from catastrophic medical expenditure.

Thank you for your comments. We have revised accordingly;

1. The use of LPGW as a threshold for affordability evaluation is taken from the WHO/HAI methodology. The treatment cost for an episode of illness is compared to the daily wage of the lowest-paid unskilled government worker to determine the number of days’ wages needed to pay for the cost of treatment. Yes, this criterion is used in all studies that use the WHO/HAI methodology. Ref. 16,17,25,26,27.

2. Shown in the MPR analysis table.

3. It shows the types of cancers that the medicine is used for. It is shown in other studies eg. Ref 26a.

4. Problems encountered are stated as a limitation.

5. The study did not collect disaggregated data showing rural and urban, as the only cancer facilities in Ghana were in the urban areas.

6. Patient out of pocket expenses are mentioned in the article.

7. Single Exit Pricing was added for consideration as a recommendation.

8. Factors included.

9. Tables presented.

10. This study used the LPGW wage as an indicator for affordability based on WHO/HAI method. The study did not use the catastrophic method, and this has been stated as a limitation

---

## [Decision Letter · Decision Letter 1]

4 Nov 2022

PONE-D-21-38310R1A Comprehensive Survey of Cancer Medicines Prices, Availability and Affordability in GhanaPLOS ONE

Dear Dr. Mattila,

Thank you for submitting your manuscript to PLOS ONE. After careful consideration, we feel that it has merit but does not fully meet PLOS ONE’s publication criteria as it currently stands. Therefore, we invite you to submit a revised version of the manuscript that addresses the points raised during the review process.

We look forward to receiving your revised manuscript.

Kind regards,

Chulaporn Limwattananon, Ph.D.

Academic Editor

PLOS ONE

Journal Requirements:

Reviewers' comments:

Reviewer's Responses to Questions

**Comments to the Author**

1. If the authors have adequately addressed your comments raised in a previous round of review and you feel that this manuscript is now acceptable for publication, you may indicate that here to bypass the “Comments to the Author” section, enter your conflict of interest statement in the “Confidential to Editor” section, and submit your "Accept" recommendation.

Reviewer #1: All comments have been addressed

Reviewer #2: All comments have been addressed

Reviewer #3: All comments have been addressed

Reviewer #4: All comments have been addressed

Reviewer #5: All comments have been addressed

2. Is the manuscript technically sound, and do the data support the conclusions?

Reviewer #1: Yes

Reviewer #2: Yes

Reviewer #3: Yes

Reviewer #4: Yes

Reviewer #5: Yes

3. Has the statistical analysis been performed appropriately and rigorously? 

Reviewer #1: Yes

Reviewer #2: Yes

Reviewer #3: I Don't Know

Reviewer #4: Yes

Reviewer #5: N/A

4. Have the authors made all data underlying the findings in their manuscript fully available?

Reviewer #1: Yes

Reviewer #2: Yes

Reviewer #3: No

Reviewer #4: Yes

Reviewer #5: Yes

5. Is the manuscript presented in an intelligible fashion and written in standard English?

Reviewer #1: Yes

Reviewer #2: Yes

Reviewer #3: No

Reviewer #4: No

Reviewer #5: Yes

6. Review Comments to the Author

Reviewer #1: The reviewer appreciates the authors efforts to address all comments and improving the manuscript. This reviewer still has some comments for improvement and hopes the authors will take these suggestion.

- Abstract - I suggest to include a sentence in the introduction paragraph describing the objective of the study. As it stands now, it is not clear and I would not suggest to have this sentence in the methods. In the results section, the sentence 'Only two cancer conditions are covered by the National Health Insurance Scheme' does not seem relevant or it is disconnected to the rest. Please revise

- The reviewer suggest to refer and include this reference also: https://doi.org/10.1186/s12913-020-05167-9.

- In the methods section you introduced the abbreviations LPG, OB, MPR, etc, could you please indicate what they mean?

- Methods - please explain why you determined the procurement efficiency. What does this measure say and why is it relevant? This measurement was not found in the results, please explain and revise.

- Methods - I would suggest to revise again how affordability was calculated (you can also refer on how it was done here: https://doi.org/10.1186/s12913-020-05167-9). It is important to justify why they calculation considered one whole month of treatment, when in reality, for most cancer treatments (mainly IV treatments) are not taken daily for a month, but once every X period of time. I strongly suggest to revise this calculations and justify better. No patient would pay for 1 month of treatment (1 treatment per day for 30 days) but would pay for 1 round of treatment (for most iv medicines, this might not be the case for tablets which might be taken daily for a month and in this case, the WHO/HAI calculation would be appropriate). Because most cancer medicines are not taken daily for a month, it might be needed to adapt WHO/HAI's calculation for this reason.

For example - having treatment from bevacizumab 400mg will only require X amount of ml from the whole vial and would only be taken once in a cycle of treatment (and not daily for a month) - although I am sure it would still be unaffordable, it would be a more fair calculation to consider how much of a medicine a patient would actually use in one round of treatment to know how (un)affordable it is. (this would also be more accurate for advocacy purposes)

- Discussion - please revise the paragraph sentences 470-481. It is difficult to understand the message you want to convey and how it relates to your results.

- Discussion - I would suggest to include the reference suggested above in line 459 and 506

- Limitations - I would also add to the affordability issue that the entire cost of treatment would be even more unaffordable considering that one course of treatment requires more than once medicine, in addition to the factors already mentioned by the authors (dispensing fees, facility fees, administration and doctors fees, etc.)

Reviewer #2: (No Response)

Reviewer #3: The authors should improve the results presentation section.

The following tables can be summarized or uploaded as supplementary files:Table 2, 5, 6 and 8.

The discussion section should show the difference between results presented in Table 3. "Percentage Availability Data using ‘mean of median’" and Figs 1a and 1b. "Percentage Availability of LPG and OB using Kruskal Wallis Test". What is new as scientific information? Is there any king of redundancy?! Just make sure to avoid duplications, cross-check the information presented in tables against the information presented in figures.

Reviewer #4: The English of the article, while understandable, does contain grammatical and spelling errors.I suggest the authors to check and correct these errors carefully.

Reviewer #5: I have only one concern. Generally, most cancer medicines are restricted drugs. So I am quite surprised to know that cancer medicines, both tablet/capsule and injection can be sold in pharmacies in Ghana. My further question is that do they need prescription? Therefore, it might be useful if authors could provide more detail on how cancer medicines are controlled and dispensed in pharmacies in Ghana.

7. PLOS authors have the option to publish the peer review history of their article (what does this mean?). If published, this will include your full peer review and any attached files.

Reviewer #1: No

Reviewer #2: **Yes: **Kofi Boamah Mensah

Reviewer #3: **Yes: **Dr. Thomas Bizimana

Reviewer #4: No

Reviewer #5: No

---

## [Author Response · Author response to Decision Letter 1]

25 Nov 2022

Reviewer 1: Thank you for your comment. We have revised the abstract text based on your feedback.

The following reference ‘Moye-Holz, D., Ewen, M., Dreser, A. et al. Availability, prices, and affordability of selected essential cancer medicines in a middle-income country – the case of Mexico. BMC Health Serv Res 20, 424 (2020). 

https://doi.org/10.1186/s12913-020-05167-9 has been referred and included.

Abbreviations have been explained in text.

Procurement efficiency removed from text, however the difference between HPM and LPM in the results gives an indication of price variations of the same medicine. This is a sign of how efficient the procurement system is in obtaining competitive prices for the medicines.

For affordability calculations, it was based on the WHO/HAI methodology (Ref 32).

We agree with you on your comments and to note that this was taken into consideration for the calculations. 

For chronic cancer illnesses, we expressed the treatment in monthly amounts, which was calculated by multiplying the daily dose by 30. For acute cancer illnesses, we considered the episodes of treatment within a month Ref: dosage column in table 8C.

The cost of a course of therapy for important conditions was compared with the daily wage of the lowest-paid unskilled government worker. This analysis is very valuable as an advocacy tool since it expresses prices in relation to an individual’s ability to pay.

Discussion: text 470-481 revised accordingly. Reference (no. 59) has been included in 459 and 506.

Limitations – text revised accordingly.

Reviewer #2: N/A

Reviewer # 3: Thank you for your comment. We have revised the text based on your feedback. 

These tables are now uploaded as supplementary files.

The percentage availability data using Kruskal Wallis test in Figs 1a and 1b is shown to complement Table 3. 

Reviewer #4: Thank you for your comments. We have revised accordingly.

Reviewer #5: Thank you for your comment. Text has been added to show that cancer medicines are only dispensed when prescribed and obtained by prescription, thus controlling its use as restricted drugs.

Editor: Thank you for your comment. This has been revised according to editors’ suggestions.

---

## [Editor Report · Decision Letter 2]

16 Dec 2022

A Comprehensive Survey of Cancer Medicines Prices, Availability and Affordability in Ghana

PONE-D-21-38310R2

Dear Dr. Phyllis Ocran Mattila,

We’re pleased to inform you that your manuscript has been judged scientifically suitable for publication and will be formally accepted for publication once it meets all outstanding technical requirements.

Kind regards,

Chulaporn Limwattananon, Ph.D.

Academic Editor

PLOS ONE

Additional Editor Comments (optional):

Line 52: xxxxx between highest-and lowest-priced should be revised to xxxxx prices.
---

## [Editor Report · Acceptance letter]

27 Dec 2022

PONE-D-21-38310R2 

A Comprehensive Survey of Cancer Medicines Prices, Availability and Affordability in Ghana 

Dear Dr. Ocran Mattila:

I'm pleased to inform you that your manuscript has been deemed suitable for publication in PLOS ONE. Congratulations! Your manuscript is now with our production department. 

Kind regards, 

on behalf of

Dr. Chulaporn Limwattananon 

Academic Editor

PLOS ONE